# Cluster-Based Aircraft Fuel Estimation Model for Effective and Efficient Fuel Budgeting on New Routes

Jefry Yanto and Rhea P. Liem *

Department of Mechanical and Aerospace Engineering, The Hong Kong University of Science and Technology (HKUST), Hong Kong SAR, China
* Correspondence: rpliem@ust.hk

**Abstract:** Fuel burn accounts for up to 25% of an aircraft's total operating cost and has become one of the most important decision factors in the airline industry. Hence, prudent fuel estimation is essential for airlines to ensure smooth operation in the upcoming financial year. Challenges arise when airlines need to estimate the total fuel consumption of new sectors where data are not available. This necessitates the derivation of a robust parametric model that can represent the characteristics of the new route even in the absence of relevant data. To address this issue, we propose a two-step approach to derive a model that can accurately estimate the aircraft fuel needed. The developed approach involves both unsupervised learning and a regression model. For the unsupervised learning step, hierarchical density-based spatial clustering of applications with noise (HDBSCAN) is used to cluster the principal component analysis (PCA)-reduced data. This step can automatically separate flight sectors based on their underlying characteristics, as revealed by their principal components, upon filtering the noise in the data. Afterward, multivariate linear regression (MLR) is used to derive the equations for each cluster. The PCA-based clustered model is shown to be superior to using a global model for a single aircraft type. This approach yields fuel estimation with less than 5% root mean square error for existing routes within each cluster. More importantly, the proposed method can accurately estimate the total fuel of a new route with less than 2% aggregate error, thereby addressing one of the current limitations in the airline fuel estimation study.

**Keywords:** aircraft fuel modeling; principal component analysis; hierarchical density-based spatial clustering of applications with noise; multivariate linear regression



## 1. Introduction

Aviation big data analytics has grown as an emerging research field in recent years [1]. Data-driven models, or machine learning techniques, have become commonly used in various industries due to the advancement in data collection and storage, and the aviation industry is no exception [2]. These techniques can help extract meaningful patterns and knowledge from any given set of data [3]. The results of previous research in aviation with these techniques have been presented in air traffic management [2], aircraft accident investigation [4], abnormality in flight operation [5], aircraft performance in airlines [6], etc.

One aircraft performance concern in airlines is fuel burn since fuel cost accounts for 17–25% of an airline's total operating expenses [7]. Having a reliable and accurate fuel estimation model is, therefore, imperative for airlines because fuel budgeting determines airlines' profitability in the following year. Fuel budgeting depends on the amount of fuel consumed and the expected fuel price. We understand that fuel price volatility is a crucial factor in fuel budgeting [8,9]. However, fuel price prediction is out of this study's scope, and we assume the airlines have appropriate strategies and policies to tackle fuel price volatility. In this study, we focus on estimating the *amount* of fuel considered for fuel budgeting purposes, which we will refer to as *fuel burn estimation* hereafter. To minimize losses due to poor fuel planning, achieving a high level of accuracy in total fuel prediction

is imperative. The airline industry spends billions of dollars on fuel, and hence, even 1% difference in accuracy translates to a substantial nominal dollar value. As an example, a local flagship carrier reported a total gross fuel cost of HKD 9.4 billion in 2021 (approximately USD 1.2 billion) [10]; 1% of this amount is around USD 12 million. Based on the discussion with our airline partner, we understand that estimating the total fuel for existing routes can be performed by relying on statistical analyses by assuming that the average fuel required is equivalent to that of the previous year's usage. A minor adjustment might be necessary for some sector-by-sector cases and is typically decided based on expert opinions and past experience. This practice usually ignores the fuel burn variance for each flight, with the assumption that the overestimation and underestimation in the fuel prediction for the individual flights will cancel each other out at the aggregate level. The validity of this assumption is illustrated in Figure 1, which shows the year-by-year variation of fuel burns for a particular aircraft type flying in different sectors. The *x*-axis shows the flight sector (i.e., origin-destination (OD) pair), and the *y*-axis shows the normalized fuel burn (note that the actual fuel burn values cannot be shown to maintain confidentiality). It is worth noting that when a different aircraft type is used, the fuel burn distributions might be different. Despite the seemingly consistent trend, this statistical approach fails when new sectors are considered due to a lack of data, as illustrated on the right-hand side of Figure 1.

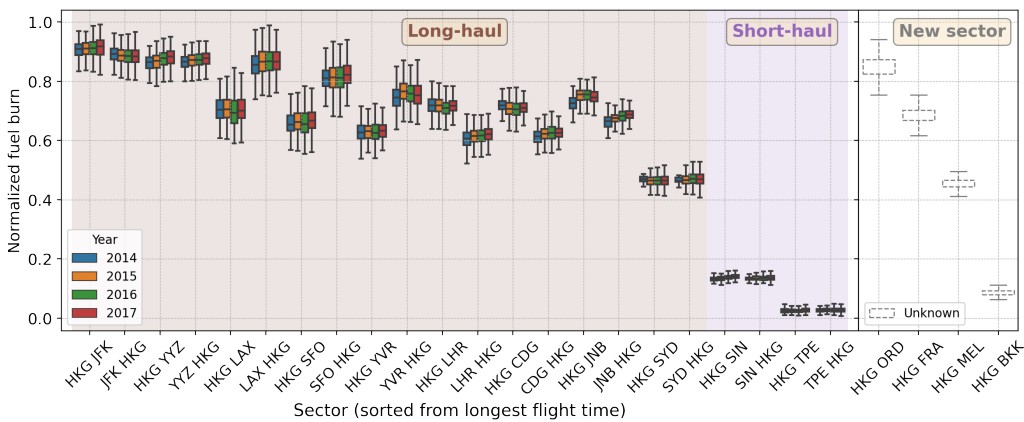

**Figure 1.** Fuel burn distributions for different sectors (from 2014 to 2017).

In the literature, there exist a wide range of fuel burn estimation models, spanning from simple models to those with higher complexity. Different types of models target different purposes, from detailed computation in one flight to lower-level computation for daily to annual operation. An overview of these models will be briefly described below, along with the reasons why they are not suitable for the specific problem at hand.

Some works were focused on representing the flight operations with as many details as possible. For instance, Lyu and Liem [11] developed a detailed flight mission analysis procedure by integrating the segment-by-segment range equation over the entire flight mission profile. For more realistic computation, the mission profile is parameterized based on actual flight data. Sun et al. [12] developed a flight simulation based on the flight equation of motion to estimate the aircraft's fuel flow during a flight. Kim et al. [13] recently developed a data-enhanced flight simulation with constraints derived from the actual operational data to mimic the actual operation closely. Lee and Chatterji [14] developed a flight simulation module with higher complexity based on the flight equation of motions that could achieve high accuracy with a takeoff weight error of less than 1%. However, all these methods require complex and detailed information within a flight. Moreover, with hundreds to thousands of flights that airlines fly annually, these methods are too computationally expensive and, therefore, unsuitable for fuel planning.

At the other end of the spectrum, some researchers derived simpler empirical models that are much less costly to compute at the expense of accuracy. This approach is more

commonly used to predict the total aggregate fuel burn, where the accuracy of fuel prediction for each individual flight is not important. O'Kelly [15] empirically derived a linear regression with respect to aircraft size and flight distance. Yanto and Liem [16] derived a multifidelity fuel burn estimation by combining low-fidelity Bréguet range equation and high-fidelity flight simulation results. This approach is especially beneficial for predicting the fuel burn of a short-haul flight, where the cruise segment is not dominant. To facilitate a simple aggregate fuel burn calculation, a linear regression model is derived for each aircraft type with the payload and flight range as input factors. Kang and Hansen [6] also developed cluster-specific ensemble learning to estimate fuel burn. They performed a stability-based K-means algorithm to cluster the United States (US) flights, and the results showed that the fuel performance of flights was direction-dependent. In the context of fuel prediction model for airlines, some researchers have performed studies on $CO_2$ emission estimation [17], fuel tankering [18], cost-to-carry [6,19], and fuel prediction for the climb out and approach segments [20]. However, to the best of our knowledge, there have not been any studies that are focused on fuel prediction modeling for airlines' fuel budgeting purposes. While there exist some models that can predict total aggregate fuel burn, as mentioned above, none of them were derived specifically for airlines' fuel budgeting purposes. First, airlines might not have the exact same set of data used in these model constructions, thereby limiting the application of these models, which are primarily data-based. Second, as mentioned earlier, the accuracy of airlines' fuel budgeting depends on fuel burn estimation and fuel price prediction, with the former being the key focus of the present paper. Predicting the total fuel consumption for a future year is challenging due to the many factors and uncertainties involved, such as air transportation demand, route variations, aircraft performance, atmospheric conditions, the introduction of new sectors, etc. As an example, an unexpected increase in the number of passengers in a particular month would result in higher fuel consumption [21]. While most of the above factors are already accounted for in the statistical approach, as previously mentioned, predicting the total fuel consumption when new sectors are introduced remains an open challenge to airlines. Hence, airlines require a robust and flexible fuel burn estimation model to ensure prudent fuel budgeting that comprehensively considers operational variations, including the introduction of new sectors. This particular capability will be one of the key contributions of the present work.

This paper presents a new fuel burn estimation approach that is also applicable for fuel prediction involving new sectors. The study uses actual operational data shared by our airline partner. The proposed approach uses a spectral theory via principal component analysis (PCA) to reduce the dimension of the available data by removing redundant information. This dimensional reduction approach helps avoid overfitting and yield more robust results. Flight clustering is then performed on the principal components by employing an unsupervised clustering algorithm. After identifying the clusters of flight sectors, a linear regression model is derived for each cluster. When a new sector emerges, we devise a means to map the new sector onto one of the clusters based on the route characteristics. The fuel burn for the new cluster can then be estimated by using the linear regression function derived for the selected cluster. The ability of the proposed method to predict fuel burn for new sectors will be assessed based on the prediction accuracy and computational complexity involved. In particular, the model needs to meet the requirements of the airlines—in this case, we take reference from our airline partner's requirement, which is to achieve less than 3% aggregate error.

This paper is structured as follows. Section 2 describes the proposed approach, including the data and methodologies used. Section 3 presents the results of the proposed approach with validation. It also discusses the performance of the proposed approach for the case of the new sectors. Section 4 concludes with a summary of the approach and results.

## 2. Methodology

We propose a systematic approach to help airlines estimate total fuel burn for fuel budgeting purposes that will be suitable for existing and new sectors. The proposed approach consists of two steps, namely an unsupervised learning with hierarchical density-based spatial clustering of applications with noise (HDBSCAN) algorithm for Step 1 and a linear regression model for Step 2. This two-step approach is illustrated in Figure 2. In Step 1 (denoted by blue arrows), we perform the HDBSCAN algorithm to group the data automatically based on the density information calculated by the algorithm. The number of clusters (three) shown in Figure 2, where each color of the dots indicates a unique cluster, are only for illustration purposes. The number of clusters obtained via performing the HDBSCAN algorithm depends on the specific problem at hand. Before we fit the data into HDBSCAN, we reduce the problem dimension through PCA to retain only input components that are dominant in determining the outputs. After data clustering, we continue to the second step. In Step 2 (denoted by red arrows), we derive a linear regression function, one for each cluster, with fuel burn as the output.

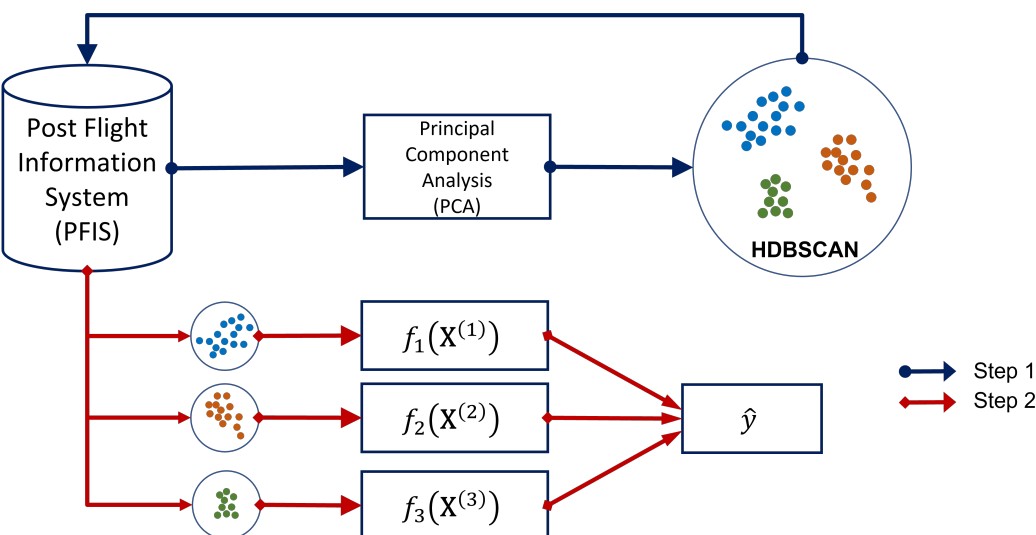

**Figure 2.** The developed two-step approach for fuel burn estimation. Note that only three clusters are shown here for illustration purposes (as indicated by the three different colors, which are arbitrarily chosen); the method is applicable for a variable number of clusters.

To predict the total fuel burn of a particular sector, we follow the same steps by first determining the cluster to which the sector (existing or new) belongs and then predicting the fuel with the corresponding linear regression model.

### 2.1. Flight Operation Data Recorded by Airlines

In this paper, we use the operation data provided by our airline partner. The data are stored in the post-flight information system (PFIS), which is recorded when the aircraft is on the ground, i.e., before takeoff and after landing. Therefore, these data are not as detailed as quick access recorder (QAR) data, which are recorded with a frequency of 1 Hz. Table 1 tabulates the input factors considered in the model derivation; these factors are identified based on our discussion with the engineers who work on fuel burn performance assessment and estimation at the airline partner. The *performance factor* feature (*Perf*) refers to the deviation in aircraft fuel performance from the optimal condition, where a negative value indicates a worse performance and a positive value indicates a better performance.

**Table 1.** List of data recorded by PFIS that are used in our study.

| Feature | Description | Unit in PFIS | Example |
|---------|-------------|--------------|---------|
| Flt Date | Flight date | YYYYMMDD | 20140101 |
| Sector | Origin destination pair | - | HKG JFK |
| AC Code | Aircraft type | - | B777-300ER |
| Flight time | Block off time | hours | 10.7 |
| Dist | Ground distance | nautical mile | 5552 |
| ZFW | Zero fuel weight | kilogram | 213,499 |
| FAF | Fuel after flight | kilogram | 8800 |
| Perf | Performance factor | % | 3 |
| Avg Wind | Average en-route wind | knots | 13 |
| ISA Dev | ISA deviation | ° C | 7 |
| MACLAW | Mean aerodynamic chord at landing weight | % | 27.06 |
| Burnoff | Fuel burn | kilogram | 100,000 |

Using flight time and ground distance information simultaneously in the model derivation may seem redundant. However, they actually complement each other. Ground distance information determines the basic range information, whereas flight time takes into account speed, climb and descent rates, altitude assignment, and holding and loitering during air traffic congestion. In other words, flight time implicitly contains information on the operational variations during flight since no flight paths are exactly the same despite sharing the same origin and destination. For the flight date feature, we only use the month information. The month variable ($m$) is transformed into a sinusoidal function, expressed as $\sin\left(\frac{2\pi m}{12}\right)$, to capture the seasonality effect. For this study, we use data pertaining to 11 origin–destination pairs or sectors inbound to and outbound from Hong Kong flown with Boeing 777-300ER and nine sectors flown with Airbus 330-300. All representative sectors are the top sectors operated by our airline partner for each aircraft type based on the total number of flights during the period of 2014–2017. Both short-haul flights (less than 6 hours) and long-haul flights (more than 6 hours) [22] are considered in the selection of representative sectors. The sectors for Boeing 777-300ER include flights to and from New York (JFK), Toronto (YYZ), Los Angeles (LAX), San Fransisco (SFO), Vancouver (YVR), London (LHR), Paris (CDG), Gauteng-South Africa (JNB), Sydney (SYD), Singapore (SIN), and Taipei (TPE). The sectors for Airbus 330-300 include flights to and from Melbourne (MEL), Perth (PER), Singapore (SIN), Kuala Lumpur (KUL), Kansai (KIX), Shanghai (PVG), Manila (MNL), Taipei (TPE), and Kaohsiung (KHH). All of these sectors will be referred to as *existing sectors* throughout the rest of the paper.

We also select some *"new sectors"* for each aircraft type to assess the efficacy of our developed solution method in achieving the intended objective. The new sectors for Boeing 777-300ER include flights to Chicago (ORD), Frankfurt (FRA), Melbourne (MEL), and Bangkok (BKK). The new sectors for Airbus A330-300 include flights to Sydney (SYD), Penang (PEN), Incheon (ICN), and Bangkok (BKK). Note that these are actually existing sectors operated by our airline partner to ensure the availability of data for validation purposes. These data, however, are strictly used to validate the effectiveness of the developed approach, and they are treated as "invisible" during the training process. The validation process will be described in Section 3.3. Using these "new sectors" to validate our model will highlight the effectiveness of our model in predicting total fuel consumption in the absence of past data.

*2.2. Spectral Decomposition*

PCA is the most widely used multivariate statistical approach, with applications in various scientific fields [23]. The objectives of using PCA are to extract essential information from data and reduce the size of the dataset by retaining only the most essential information. The principal components (PCs) are constructed with several orthogonal basis functions, which are essentially linear combinations of the original bases. Depending on the required level of accuracy and available computational budget, users can determine the number of

PCs to represent the dataset. PCA works better when the data are within a similar range, which can be achieved by centering the mean to zero and dividing them by the standard deviation. First, we perform a singular value decomposition (SVD) analysis on our data $\mathbf{X}$ of $n$ samples and $m$ features into a desired new dimension $N_d$ as follows:

$$\mathbf{X} = \mathbf{LDR}^{\mathrm{T}}, \tag{1}$$

where $\mathbf{L} \in \mathbb{R}^{n \times N_d}$ contains the left singular vectors, $\mathbf{R} \in \mathbb{R}^{N_d \times m}$ contains the right singular vectors, and $\mathbf{D}$ is the diagonal matrix of singular values. Note that $\mathbf{D}^2$ is the diagonal matrix of the (nonzero) eigenvalues of $\mathbf{X}^{\mathrm{T}}\mathbf{X}$ and $\mathbf{X}\mathbf{X}^{\mathrm{T}}$. We can then find the PC ($\mathbf{P}$) projection as follows:

$$\mathbf{P} = \mathbf{LD} = \mathbf{XR}. \tag{2}$$

More detailed explanations about PCA can be found in [23].

### 2.3. Clustering Algorithm

Campello et al. [24] proposed HDBSCAN as an extension of density-based spatial clustering of applications with noise. Both methods were derived with a density-based clustering technique. HDBSCAN can improve the performance of clustering results by introducing the hierarchical clustering method. In HDBSCAN, we do not need to predefine the number of clusters since the algorithm will automatically define the threshold for each cluster based on the density. More detailed explanations about HDBSCAN can be found in [25]. Python language programming users can also access the provided library through https://hdbscan.readthedocs.io/ (accessed on 16 October 2022). HDBSCAN has been successfully applied in several real-world problems such as trajectory clustering [26,27], movement recognition [28], text recognition [29], meteorological prediction [30], etc.

### 2.4. Regression Analysis

A multivariate linear regression model is a linear equation used to represent a relationship between input variables $\mathbf{X} \in \mathbb{R}^{n \times m}$ and output variable $\mathbf{Y} \in \mathbb{R}^n$ [31]. Let us express $\mathbf{Y}$ and $\mathbf{X}$ as:

$$\mathbf{Y} = \begin{bmatrix} y_1 \\ y_2 \\ \vdots \\ y_n \end{bmatrix}_{n \times 1} \quad \mathbf{X} = \begin{bmatrix} x_{11} & \cdots & x_{m1} \\ x_{12} & \cdots & x_{m2} \\ \vdots & \vdots & \vdots \\ x_{1n} & \cdots & x_{mn} \end{bmatrix}_{n \times m}, \tag{3}$$

where $n$ and $m$ are as previously defined; generally, $n > m$. The multivariate linear regression model can then be expressed as:

$$\mathbf{Y} = \mathbf{B} + \mathbf{XC} + \mathbf{\Delta}, \tag{4}$$

where $\mathbf{B} \in \mathbb{R}^n$ contains the bias terms, $\mathbf{C} \in \mathbb{R}^m$ contains the coefficients of the linear equation, and $\mathbf{\Delta} \in \mathbb{R}^n$ corresponds to the residual term. By solving the least square to minimize $\mathbf{\Delta}$, we can solve for $\mathbf{B}$ and $\mathbf{C}$. This function will be used for the regression model in this paper.

## 3. Results and Discussion

In this section, we present the results of the developed parameterized fuel burn model and compare them against existing approaches. For comparison purposes, we derive a linear regression for each aircraft type following our previous work [16]. We will refer to this model as the *global model* hereafter. We split the available data into a training set and a test set. We randomly draw 800 flight samples from each sector from the year 2014–2016 for the training set, and the test set comprises flights from the year 2017. The number of samples is determined based on the minimum number of flights available for each sector.

This sampling helps ensure a more equal distribution of data across all sectors to avoid overfitting in a particular sector. Figure 3 shows the global model prediction performance on the test set for existing and new sectors, which were listed in Section 2.1. The *x*-axis shows the sectors sorted from the longest to shortest flight time (for outbound and inbound flights), and the *y*-axis shows the root mean square error (RMSE) in kilograms. The global model shows the RMSE for Boeing 777-300ER up to 3300 kg and up to 2000 kg for Airbus 330-300. Similar error ranges are also observed in the new sectors.

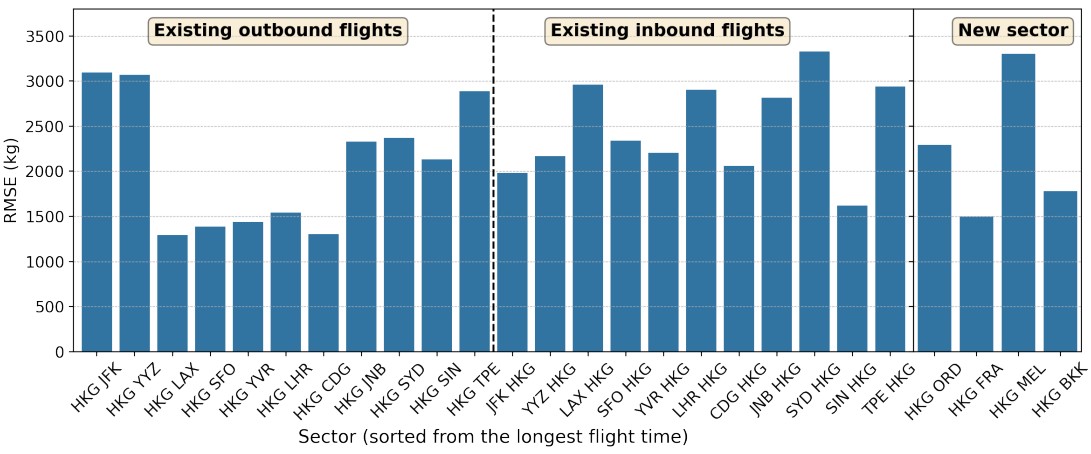

(**a**) Boeing 777-300ER

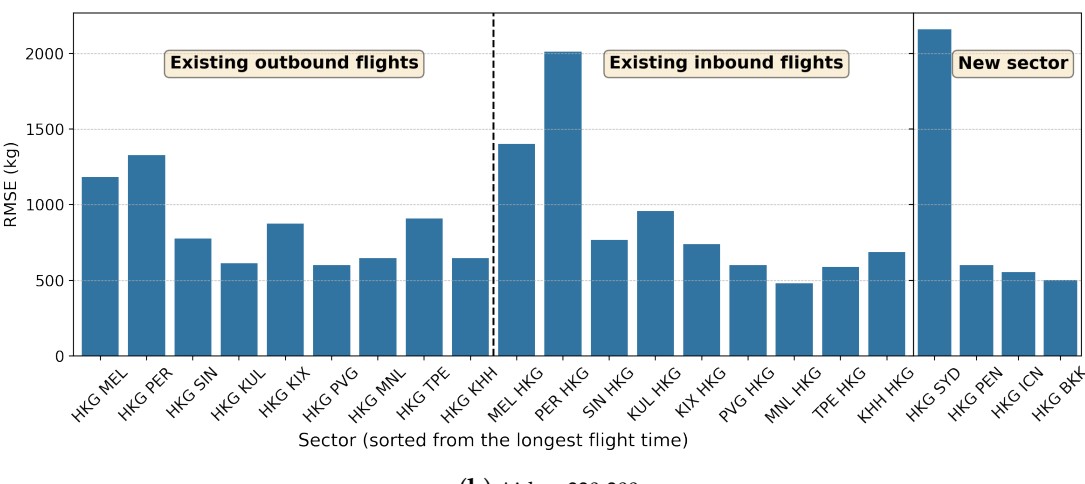

(**b**) Airbus 330-300

**Figure 3.** Global model prediction performance on the test set, separated for existing outbound and inbound flights and the new sector. The sectors are sorted from the longest to shortest flight time.

The lack of trend observed in Figure 3 for both aircraft types considered in this study suggests that each sector has a unique fuel burn behavior. As such, a *sector-by-sector model*, where a linear regression is derived for each sector, might be necessary to capture the uniqueness of each sector. However, this sector-by-sector model can not be used to predict the total fuel burns of new sectors due to the lack of sector data and the appropriate regression model. When this situation arises, we can select another "similar" sector as a reference and use its regression function, but this approach is too subjective. In this paper, we aim to devise a more systematic method to pick the appropriate regression function for the new sector, thereby eliminating subjectivity. We explain this systematic method, namely the *PCA-based clustered model*, and our developed model performance in Section 3.1. For a more comprehensive comparison and model assessment, another cluster-based linear regression model, namely the *naïvely clustered model*, is derived and demonstrated with the

same datasets. This *naïve clustering* is based solely on the distance/flight time as suggested by the International Air Transport Association (IATA) [22]. In other words, the naïve clustering process does not follow any systematic approach. Table 2 summarizes the four regression models considered in this study.

**Table 2.** Summary of regression models used for fuel burn prediction and their usage with new sectors.

| Model | Definition |
|---|---|
| Global model [16] | One regression model is derived for each aircraft type, to be used for all sectors (including new ones) |
| Sector-by-sector model | One regression model is derived for each combination of aircraft type and flight sector; fuel prediction for a new sector relies on a subjective selection of a "similar" sector |
| Naïvely clustered model | One regression model is derived for each combination of aircraft type and cluster, where the cluster is naïvely defined following the definition by IATA; fuel prediction for a new sector uses the regression model for the relevant cluster |
| PCA-based clustered model | One regression model is derived for each combination of aircraft type and cluster, where the cluster is defined upon performing the PCA-based HDBSCAN clustering (as illustrated in Figure 2); fuel prediction for a new sector uses the regression model for the relevant cluster |

A deeper understanding of our developed model through sensitivity analysis will be presented in Section 3.2. Recall that one of the key capabilities and contributions of the proposed method is to predict the total fuel consumption of a new sector where data are not available. We will demonstrate this capability in Section 3.3.

*3.1. PCA-Based Clustered Model*

The developed PCA-based clustered model represents a more systematic, robust approach that is suitable for predicting total fuel consumption of new sectors, thanks to its ability to systematically identify and quantify the underlying characteristics of flight sectors and use this information in model derivation. In this section, we demonstrate our approach (as explained in Section 2) for both aircraft types. The clustering results (Step 1) for Boeing 777-300ER flights are shown in Figure 4a. To illustrate the data in a two-dimensional space, we use the uniform manifold approximation and projection (UMAP) (https://umap-learn.readthedocs.io/(accessed on 16 October 2022)). The two axes represent the location of data points in the manifold upon performing the projection algorithm, where the *x*-axis is the first manifold axis, and the *y*-axis is the second manifold axis. The tickers are not shown in plots for simplicity since the exact values of manifold projections are not used in our analyses. The left figure shows the original data, with each color representing a different sector. The data are scattered and create small clusters for each sector. In other words, the patterns of similar characteristics among different clusters are not yet revealed in this plot. Once we perform PCA on the data, three distinct clusters emerge, as shown in the middle figure (the data points are still color-coded based on their sectors). HDBSCAN also automatically labels these clusters into three different clusters, as shown with different colors in the right figure. We can see that the PCA and HDBSCAN separate the long-haul sectors into two. We distinguish these two long-haul clusters as long-haul AU (color-coded in orange) for intercontinental flights to Australia (this will be called *LH-AU B773ER*) and long-haul US *and* EU (color-coded in green) for intercontinental flights to America and Europe (this will be referred to as *LH-US-EU B773ER*). The short-haul sectors are grouped and named short-haul (color-coded in blue), which will be referred to as *SH B773ER*. To better illustrate the clustering method results, these color-coded flights are shown on the world map in Figure 4b; the new sectors (in red) are also shown for illustration purposes. Our proposed approach yields clusters that are different from those obtained from performing naïve clustering, which only categorizes flights as short-haul and long-haul.

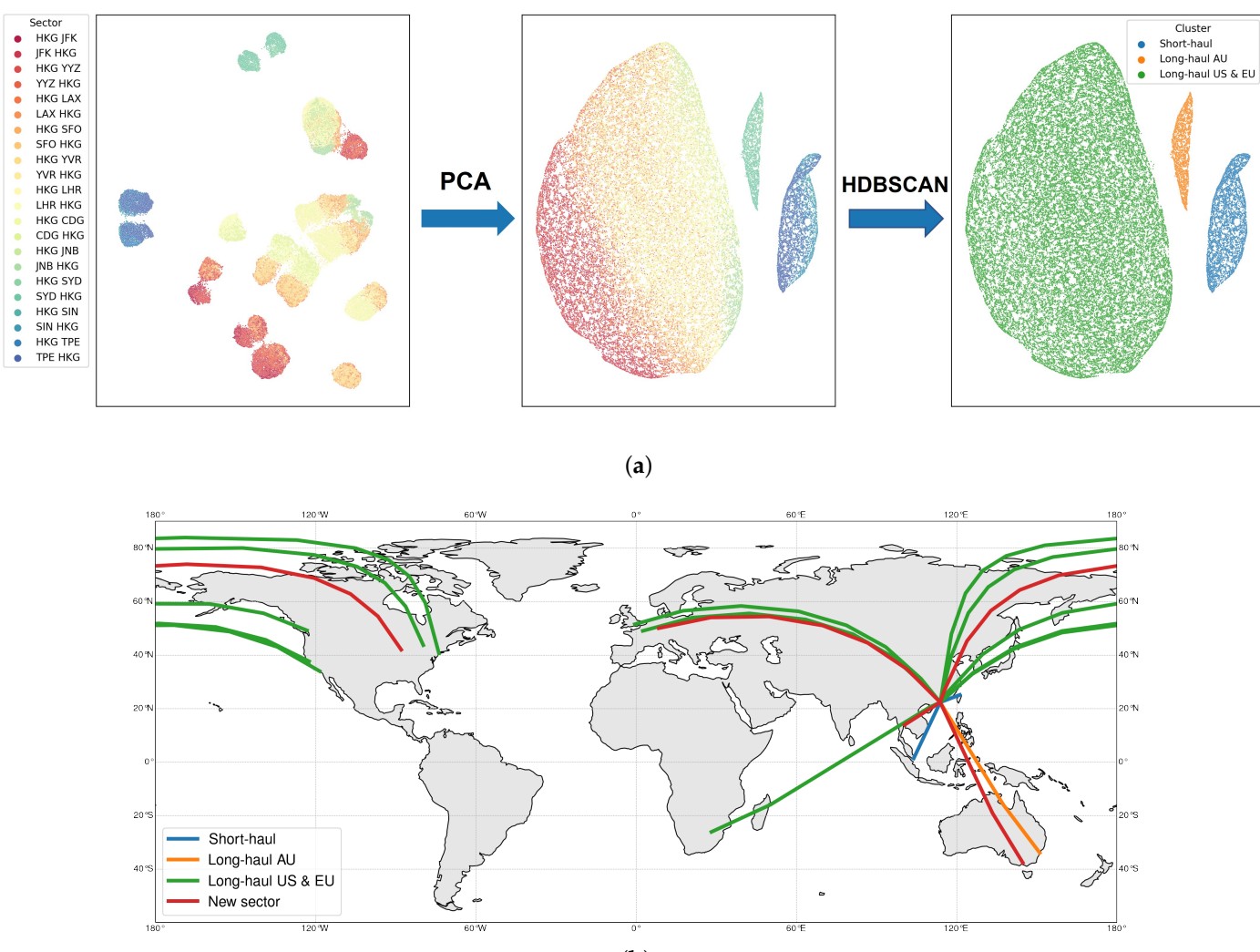

**Figure 4.** Visualization of the proposed approach for aircraft type Boeing 777-300ER. (**a**) The left figure shows the data before PCA. The middle figure shows the data clustered into three groups after PCA. The right figure shows the HDBSCAN clustering results. (**b**) Clusters of the sectors in the world map, including the new sectors.

Figure 5a shows similar clustering results for Airbus 330-300 flights. The sectors are now divided into two clusters, namely the long-haul AU (color-coded in pink) for intercontinental flights to Australia (*LH-AU A333*) and the short-haul flights (color-coded in olive) (*SH A333*). As before, the projected flight sectors on the world map and their cluster identifications are shown in Figure 5b.

The two results presented above show that intercontinental flights to Australia have a unique characteristic since they are automatically separated from other flights for both aircraft types. This shows that in addition to flight range, the geographical location of the origin or destination can also affect flight characteristics and fuel burn performance. Recall that Australia has the opposite seasonal characteristics from countries in the Northern Hemisphere. The comparison between the PCA-based and naïve clustering results, which will be presented in Section 3.3, will reveal the importance of identifying flight characteristics beyond distance-based characterization.

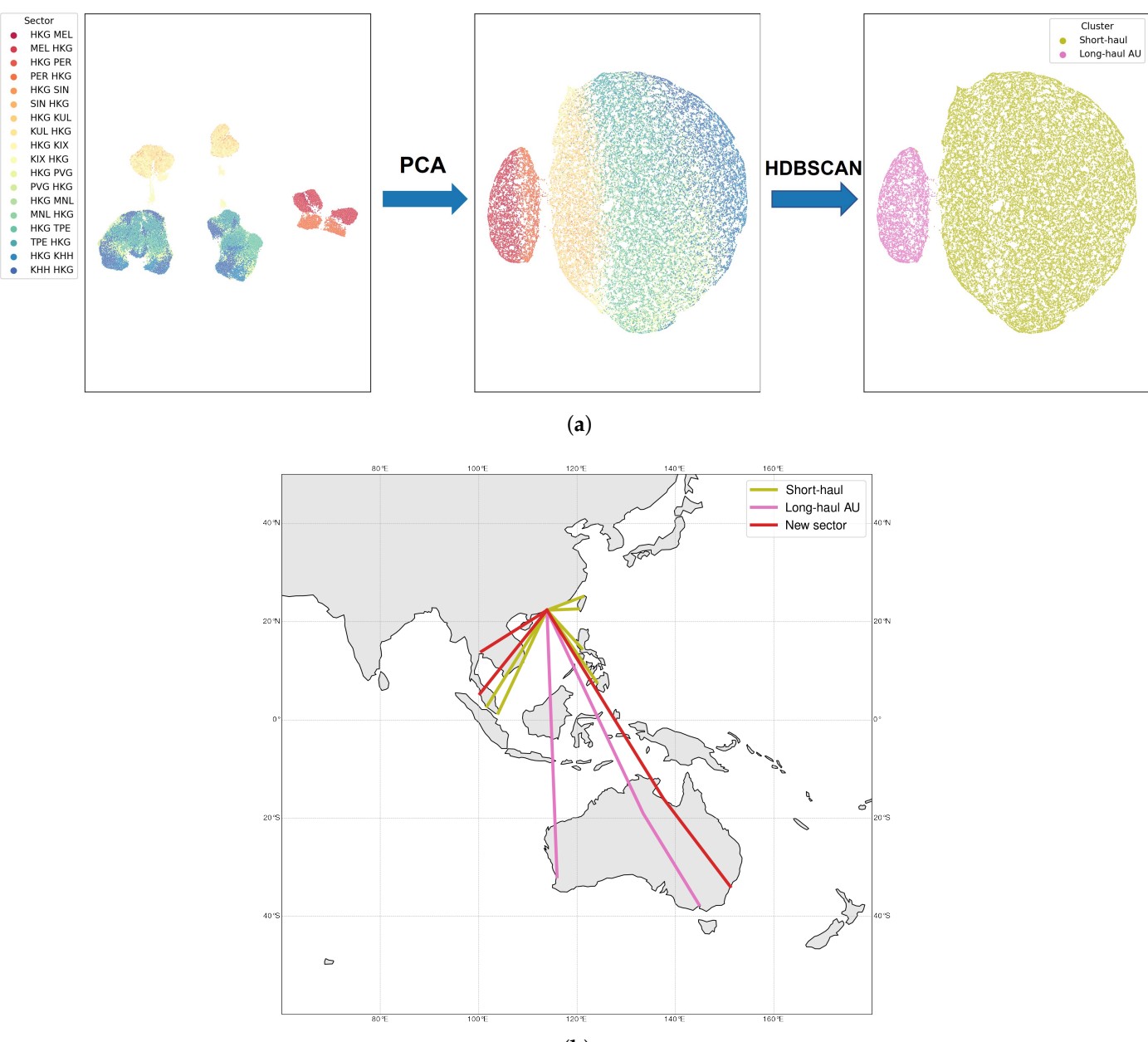

**Figure 5.** Visualization of the proposed approach for aircraft type Airbus A330-300. (**a**) The left figure shows the data before PCA. The middle figure shows the data clustered into three groups after PCA. The right figure shows the HDBSCAN clustering results. (**b**) Clusters of the sectors in the world map, including the new sectors.

For Step 2, we derive the linear regression for each cluster defined by HDBSCAN, and these models will be referred to as the *PCA-based clustered model* hereafter. The test sets that we set aside earlier are used to validate the models; the results are shown in Figure 6. The figure shows the error in terms of RMSE (in kilograms and as a percentage) and an aggregate error (as a percentage). All RMSE values are below 1500 kg and 5%. Different trends are observed in the RMSE values expressed in kilograms and as a percentage. While the RMSE in kilograms is smaller for short-haul flights, the RMSE as a percentage is larger. The latter is due to the lower reference fuel consumption used to calculate the percentage; since short-haul flights burn less fuel, the percentage is higher even though the nominal

error is lower. All aggregate errors fall below 1.5%, which meets the airline's requirement (below 3%), as mentioned in Section 1.

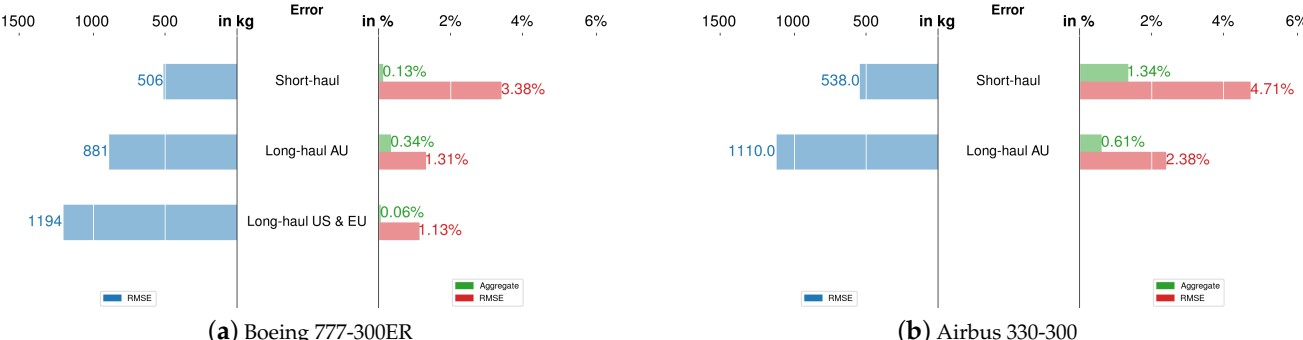

(**a**) Boeing 777-300ER　　　　　　　　　　　　　　　(**b**) Airbus 330-300

**Figure 6.** Validation of the PCA-based clustered model. The left figure shows the error (RMSE) in kilograms and the right figure shows the error (aggregate and RMSE) as a percentage.

To evaluate and compare the performance of the new PCA-based clustered model against other models listed in Table 2 (i.e., global model, sector-by-sector model, and naïvely clustered model), we train all models using the same training set and compute the RMSE for the same test set. Figure 7 shows the performance of these four approaches on the test set, with each sector on the *x*-axis (sorted from the longest to shortest flight time) and RMSE (in kilograms) on the *y*-axis. The new PCA-based clustered model is shown to perform better than the global model for both aircraft types. Our PCA-based clustered model also performs slightly better than the naïvely clustered model for the long-haul flight of the Boeing 777-300ER. Moreover, our PCA-based clustered model offers an additional benefit, it can systematically predict the total fuel burn of a new sector, which will be demonstrated in Section 3.3. These results highlight that deriving a global fuel burn estimation model for each aircraft type is insufficient to represent different flight characteristics (based on flight range and geographical location). In addition, deriving a different model for each sector would be overkill since the added complexity does not bring any significant improvement in terms of estimation accuracy. Cluster-based regression models offer a balanced compromise between using the global model and the sector-by-sector model. The benefits of using our PCA-based clustered model compared to the naïvely clustered models—especially for predicting total fuel consumption of new sectors—will be discussed in Section 3.3.

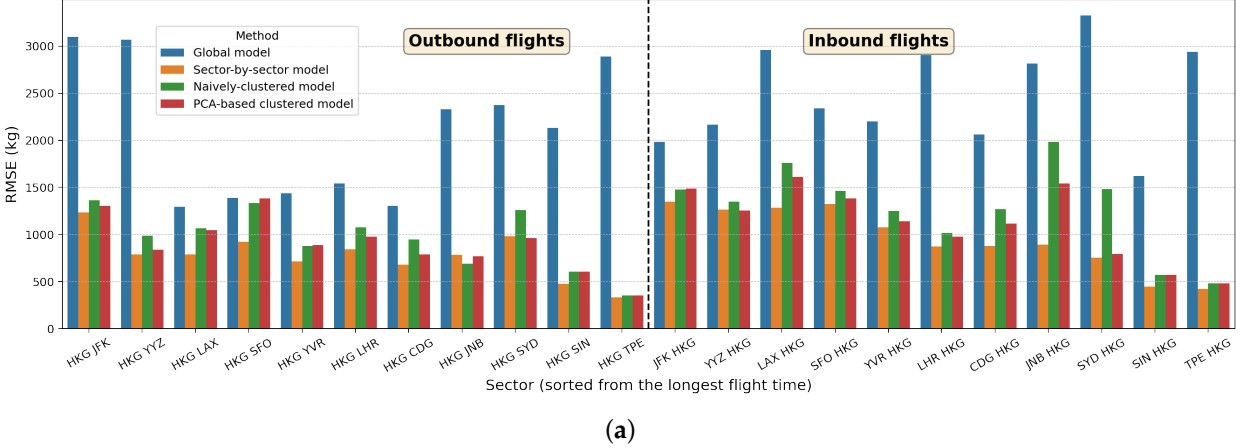

(**a**)

**Figure 7.** *Cont.*

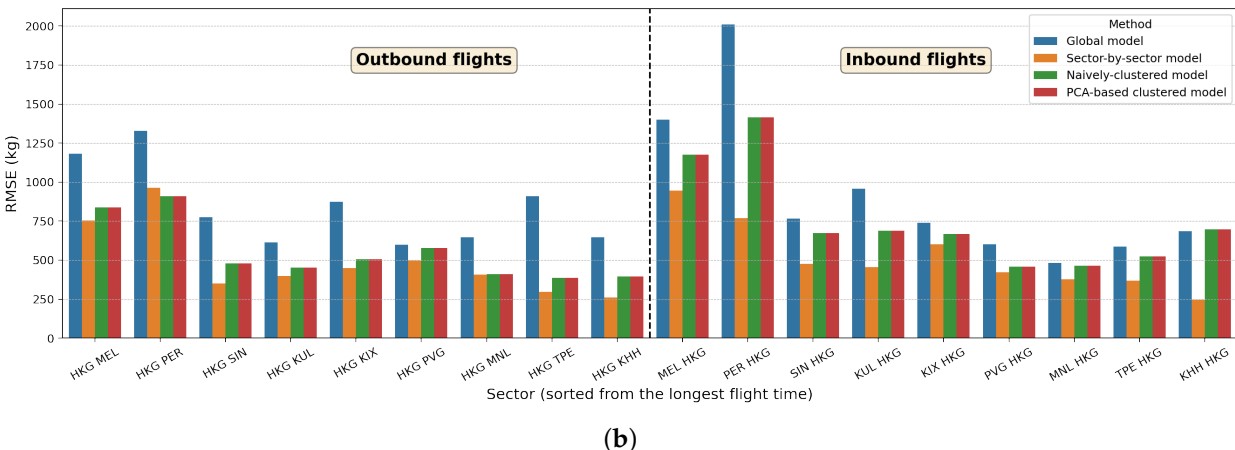

(**b**)

**Figure 7.** Performance of four different linear regression models on the test set for each sector, both on (**a**) Boeing 777-300ER and (**b**) Airbus 330-300.

We also perform sensitivity analyses to identify the dominant input factors within each cluster. The results, which show the uniqueness of each cluster, will be presented in Section 3.2. The sensitivity analysis study results provide important insight into the significant features required to predict fuel burn.

### 3.2. Sensitivity Analysis

Generally, sensitivity analysis is used to study the contribution of each input feature to the output [32]. For parametric regression models, the regression coefficients provide information that can represent the model's sensitivity. Figure 8 shows the coefficient of linear regression on the $y$-axis and cluster identifier on the $x$-axis; one plot for each input factor. In each plot, the grey bars represent the regression coefficients for the input factor of interest (e.g., flight time for the top-left plot), whereas the coefficients for PCA-based clustered models are identified by the same colors as those used as cluster identifiers in Figures 4 and 5.

The sector-by-sector coefficients are similar within each cluster except for MACLAW and month features. This behavior indicates that the features are weaker than others to represent fuel burn. The cluster's coefficients are shown to have similar magnitude and direction to the mean of the sectors' coefficients. However, we notice different magnitudes and directions (when comparing the sector's coefficient and cluster's coefficient) for some features. Note that in this context, a higher coefficient does not necessarily mean a more notable contribution to the output because the result still depends on the unit and value of the feature. For instance, $y = ax$ with $a = 1000$ and $x$ in grams is not more significant than $a = 1$ and $x$ in kilograms since the result will still be the same. Therefore, further analysis is required to understand the contribution of each feature to the fuel burn.

In order to study the contribution of each feature in the clustered model to the fuel burn, we calculate the mean contribution as a percentage as follows:

$$\%\text{Mean contribution} = 100 \times \frac{\sum_{j=1}^{n} \frac{x_{ij} \times c_{ij}}{\hat{y}_j}}{n}, \quad i = 1, ..., m. \tag{5}$$

The mean contribution quantifies the averaged proportion of fuel contributed by one regression term, $x_{ij} \times c_{ij}$ (corresponding to one particular feature), to the total fuel for the flight. The mean contribution is presented in Table 3 with the coefficient of the clustered model. The bold number indicates the contribution of the corresponding feature to the fuel burn, which is more than 50%, indicating the importance of the input feature. Meanwhile, the features with less than 10% contribution are considered less important.

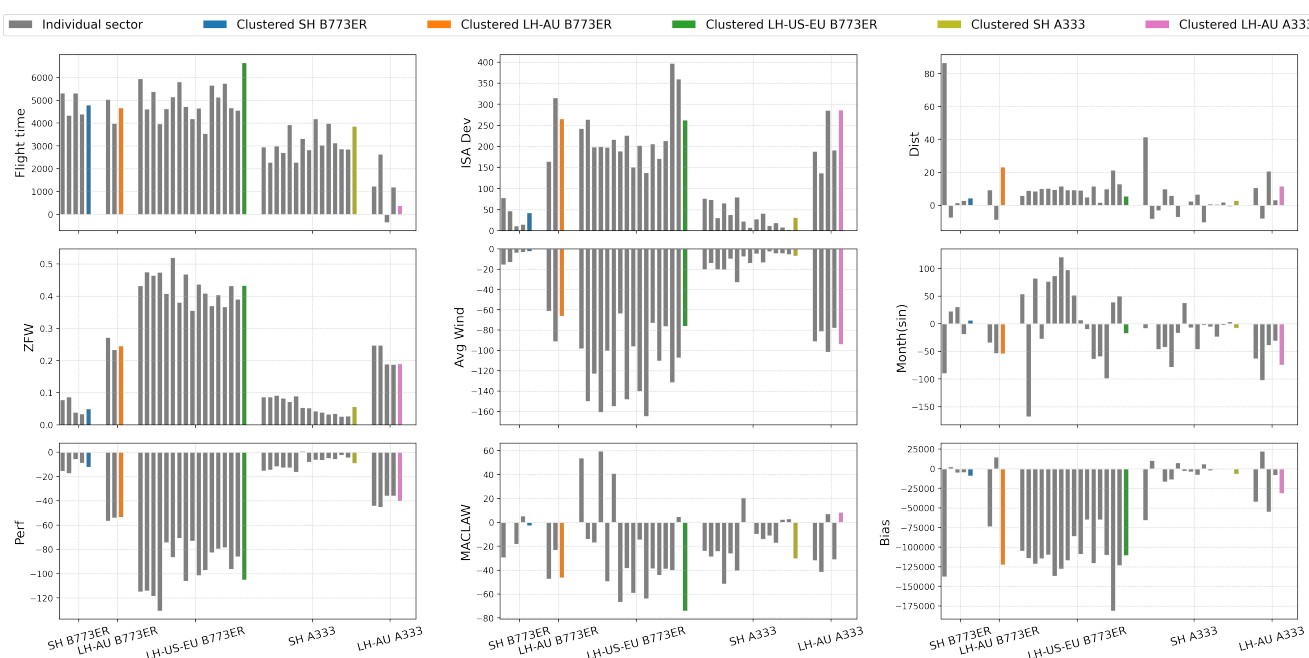

**Figure 8.** Comparison of the linear regression coefficient for two different aircraft types (Boeing 777-300ER and Airbus 330-300) for different clusters.

**Table 3.** Coefficients for derived cluster-based linear regression models for Boeing 777-300ER and Airbus 330-300. The mean contributions to the total fuel are also presented as percentages.

| Cluster | Coefficient of Linear Regression (% Mean Contribution to Fuel) | | | | | | | | |
|---|---|---|---|---|---|---|---|---|---|
| | **Flight Time** | **ZFW** | **Perf** | **ISA Dev** | **Avg Wind** | **MACLAW** | **Dist** | **Month (Sin)** | **Bias** |
| SH B773ER | **4809** (**63**) | **0.05** (**84**) | −12 (1.5) | 43 (2.8) | −2.65 (0.02) | −2.86 (−0.62) | 4.5 (23) | 6.45 (0.0) | −9289 (−**75**) |
| LH-AU B773ER | **4683** (**59**) | **0.25** (**79**) | −54 (1.1) | 266 (4.3) | −66 (−0.01) | −46 (−1.8) | **23** (**140**) | −55 (0.0) | −**122,823** (−**182**) |
| LH-US-EU B773ER | **6650** (**83**) | **0.43** (**91**) | −105 (1.5) | 263 (0.8) | −76 (−1.4) | −74 (−1.9) | 5.7 (35) | −18 (0.0) | −**110,718** (−**108**) |
| SH A333 | **3873** (**63**) | **0.05** (**83**) | −9.2 (0.3) | 31.5 (2.5) | −7 (−0.07) | −30.4 (−7.3) | 3.07 (20) | −8.2 (0.0) | −6723 (−**63**) |
| LH-AU A333 | 398 (6.7) | **0.19** (**63**) | −40 (−0.22) | 287 (4) | −94 (0.1) | 8.5 (0.5) | **11.7** (**90**) | −74 (0.0) | −**31,560** (−**64**) |

The dominant input features can reveal insight into flight-sector characteristics, as described briefly below. We notice that the Boeing 777-300ER fuel burn depends primarily on flight time, followed by distance, except for long-haul AU flights. This observation is also consistent with long-haul AU flights flown with Airbus 330-300, which suggests that flights to Australia tend to have relatively more similar speed profiles and consistent flight routes, as reflected in the less dominant impact from flight time. Another interesting finding is that for short-haul flights flown with both aircraft types, zero-fuel weight is found to be significant. Note that zero-fuel weight is highly correlated with takeoff weight, which determines the fuel consumption. This makes sense since short-haul flights have a shorter cruise range but a more dominant climb segment. The fuel consumed during climb highly depends on the aircraft's weight and contributes significantly to the total fuel for short-haul flights [11]. The results match our intuition that flight time, zero-fuel weight, and distance are the most significant features. The varying coefficients of significant features for different

clusters further highlight the insufficiency of a global model (which uses the same set of model coefficients for all sectors) in representing different flight-sector characteristics.

*3.3. Prediction in New Sectors*

A new sector is defined as an origin-destination pair without any historical data. For route planning and preliminary analysis purposes, most airlines use a flight planning system (FPS) that can provide reasonable aircraft information (e.g., aircraft weight and performance factor), route information (e.g., flight time and distance), and atmospheric condition information (e.g., wind). The resulting predictions, however, are not very accurate because a large part of the operational variations is not accounted for. Therefore, airlines cannot rely on FPS projection alone in estimating the fuel burn of a new route.

In this study, we use some existing sectors to emulate new sectors, apart from those used to train the models, as mentioned in Section 2. By doing so, we can properly validate the models, thanks to the available fuel consumption information. The model validation is performed by assessing RMSE and the aggregate error, following their definitions and usage in Section 3.1. In addition, we also compare the results obtained from our PCA-based clustered model with those from the global model and naïvely clustered model.

To predict fuel burn using our PCA-based clustered model, the first step is to define the appropriate cluster for the new sector. When the sector characterization within each cluster is well defined (e.g., long-haul to the Southern Hemisphere vs. long-haul to the Northern Hemisphere vs. short-haul), it is reasonable to assign the cluster manually. The HKG-MEL sector, for instance, can be classified into a long-haul AU cluster due to its similarity to HKG-SYD. Once the cluster is defined, we can simply use the corresponding regression model using inputs pertaining to the new sector generated by FPS. Figure 9 shows the comparison of three different models, namely the global model, naïvely clustered model, and the PCA-based clustered model, for two aircraft types. The results show clear advantages of using the clustered models as compared to the global model, which shows that a global model is not sufficient in representing the sector variations in aircraft operations. In this study, Boeing 777-300ER flights have more diverse sectors (with three PCA-based clusters) compared to Airbus 330-300 (with only long-haul and short-haul clusters). Recall that the naïvely clustered model is determined based only on the flight range (short-haul vs. long-haul), with no distinction on the geographical direction of the origin/destination. Hence, it is not surprising that both PCA-based and naïvely clustered models yield the same prediction results and accuracy for Airbus 330-300 flights. The superiority of the PCA-based clustered model is demonstrated in the fuel burn prediction for the "new" HKG-MEL sector, where reductions in error metrics can be clearly observed. This is because the PCA-based clustering procedure can automatically identify the different characteristics between different long-haul flights (US and EU vs. AU). In this study, the demonstration of the benefits of our developed approach is limited due to data limitations. When more sectors (with more distinction in their characteristics) are considered in the training and clustering procedures, the benefits of the systematic and automatic PCA-based clustering approach will be more prominent.

One might wonder about the applicability of our approach when the availability of data (in terms of the number of variables) is not as extensive as what airlines can provide. The generated FPS inputs are often confidential and limited to the airlines. For this reason, we also derive *simplified models* that are constructed with only three inputs, i.e., flight time, zero-fuel weight, and flight distance. These three inputs are identified as the most dominant features by the sensitivity analysis, as shown in Table 3. Recall that flight time and flight distance features are complementary to each other; the relationship between the two cannot be simply described by means of velocity due to the three-dimensional nature of flight trajectories. Two flights can fly the same distance but with different flight times owing to variations in aircraft speed, altitude, congestion, or any air traffic disruption. For comparison purposes, we label the models with fewer features as the *simplified global model* and *simplified PCA-based clustered model*. These simplified models use the same

regression coefficients as those listed in Table 3, but only consider the bias term and three inputs obtained from flight information while setting the rest of the inputs to their nominal values as listed in Table 4. Note that these nominal values are used for both aircraft types, i.e., Boeing 777-300ER and Airbus 330-300. This is to simulate the situation when the parametric model is available, but users have no access to detailed flight information inputs. We then analyze the original global and PCA-based clustered models with the simplified global and PCA-based clustered models' performance; the results are shown in Figure 10.

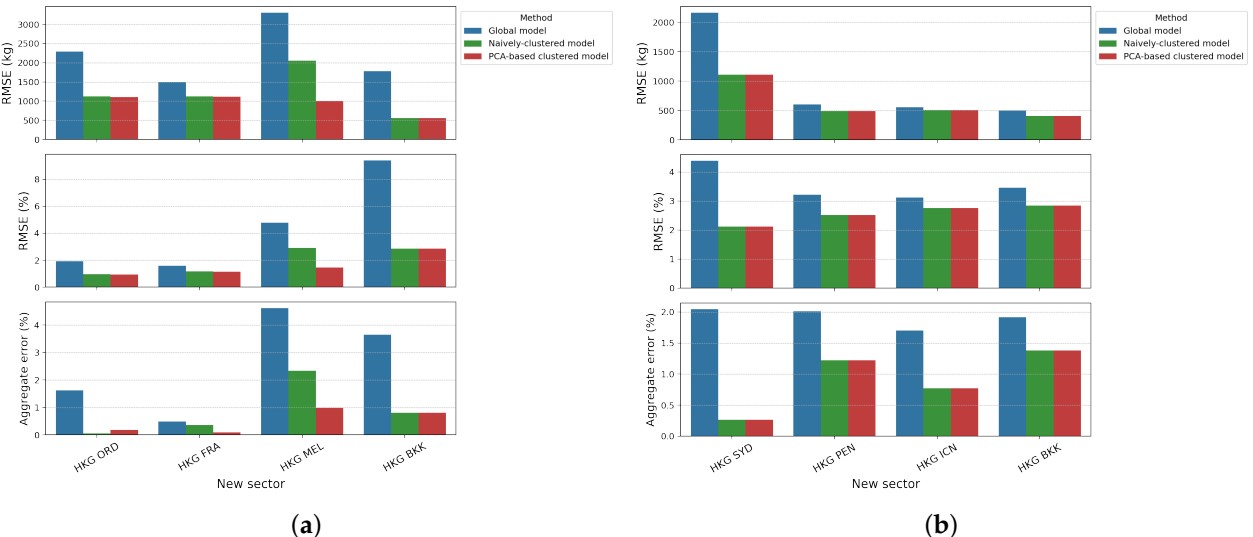

**Figure 9.** Comparison of fuel prediction performance of the global model, naïvely clustered model, and PCA-based clustered model to a new sector both for (**a**) Boeing 777-300ER and (**b**) Airbus 330-300. The top graph shows RMSE values in kilograms, the middle graph shows the RMSE values as percentages, and the bottom graph shows the aggregate error as a percentage.

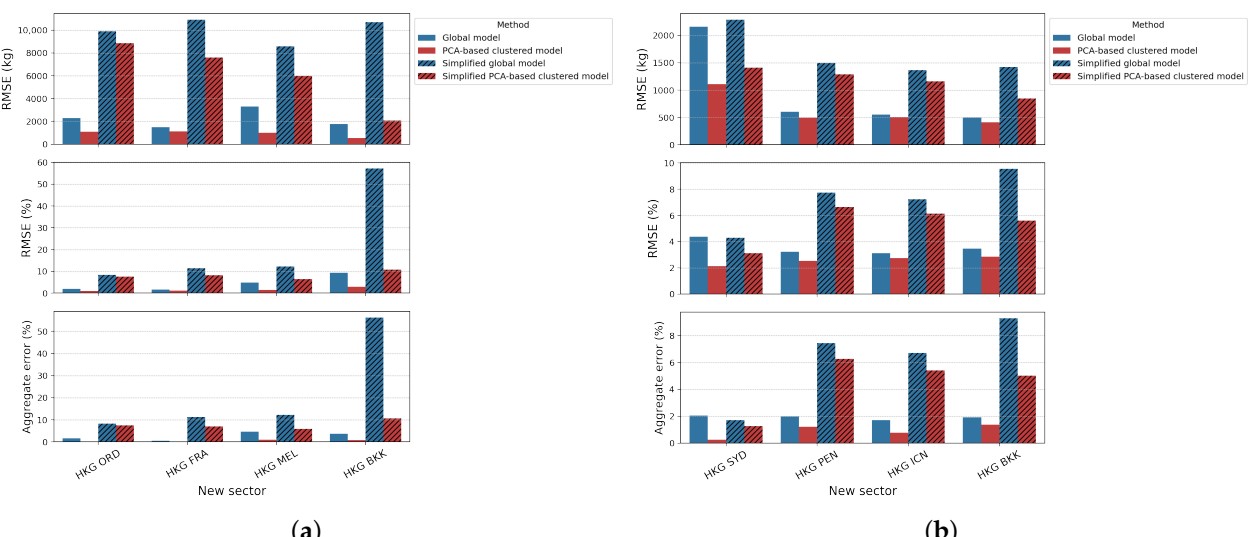

**Figure 10.** Comparison of fuel prediction performance of the original global model, original PCA-based clustered model, simplified global model, and simplified PCA-based clustered model to the new sector for (**a**) Boeing 777-300ER and (**b**) Airbus 330-300. The top graph shows the RMSE values in kilograms, the middle graph shows the RMSE values as percentages, and the bottom graph shows the aggregate error as a percentage.

**Table 4.** Nominal values for input factors that are "excluded" in the simplified global model and simplified PCA-based clustered model for both Boeing 777-300ER and Airbus 330-300.

| Feature | Nominal Value |
| --- | --- |
| Perf | 0 |
| Avg Wind | 0 |
| ISA Dev | 0 |
| MACLAW | 20 |
| Month (sin) | 0 |

The results show that the simplified PCA-based clustered model can still offer an acceptable accuracy. While all simplified global models show higher errors than the corresponding simplified PCA-based model, Figure 10a shows a notably larger difference in accuracy between the two models for the HKG-BKK sector. Further investigation reveals that the deviation of the actual average performance factor ($-16\%$) from the assumed nominal value (0%, as shown in Table 4) contributes to the larger error. The *Perf* coefficient for the global model is $-84$, whereas it is $-12$ for the PCA-based clustered model (as shown in Table 3). Hence, the difference between the predictions of the global model and the simplified global model ($-84 \times Perf$) is more substantial than that of the PCA-based clustered model and the simplified one ($-12 \times Perf$). The importance of choosing the appropriate nominal value for the simplified model is illustrated in Figure 11, which shows a decrease in RMSE when the nominal *Perf* value is closer to the actual average value. Note that the large discrepancy is not observed in other sectors since their average *Perf* values (ranging between $-5\%$ and $5\%$) are closer to the assumed nominal value.

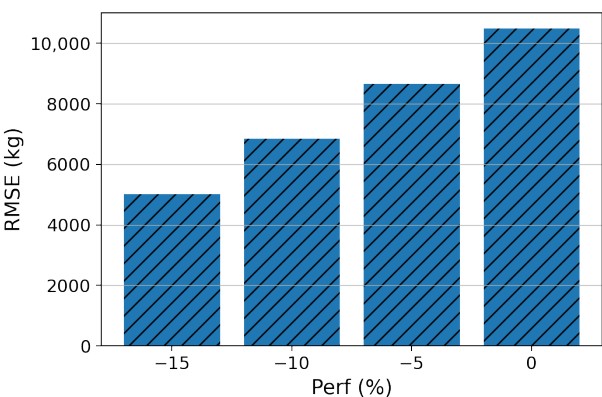

**Figure 11.** RMSE values of the simplified global model for HKG-BKK with a different nominal value of *Perf*, which shows the improvement in accuracy when the nominal *Perf* value is closer to the actual average ($-16$).

The above results demonstrate that the developed PCA-based clustered model consistently performs better than the global model even when some real feature values are missing. Moreover, this simplified PCA-based clustered model is easier to implement by other users since only basic flight information is required as inputs.

### 4. Conclusions

We have demonstrated a new approach to estimating fuel burn that will be useful for airlines' budgeting purposes since we use information and data that are specifically available to airlines. One of the key objectives is to develop an approach that can systematically predict the total fuel consumption of new flight sectors, in addition to the existing ones, thereby addressing the limitation of existing models. In particular, a PCA-based clustered model has been developed. The developed approach was implemented for two aircraft types, namely Boeing 777-300ER and Airbus 330-300. The results showed that it is

necessary to derive a fuel model for each aircraft type due to the different fuel performance characteristics; these results agree with our previous observations [16]. Moreover, our study showed that a global linear regression model for each aircraft type, which is commonly adopted in other fuel models, is insufficient. This is due to the fact that fuel performance across different sectors is not homogeneous. In other words, how much fuel is burned does not only depend on how far the aircraft flies but also on the mission profile and where it flies. Short-haul and long-haul flights have different fuel burn characteristics owing to the different dominant segments in different types of flights. The climb segment dominates fuel consumption in short-haul flights, whereas the cruise segment dominates in long-haul flights [11]. Therefore, this differentiation needs to be considered in model derivation, which we did with the PCA-based clustered model. Furthermore, using a cluster-based regression model eliminates the need to derive a sector-by-sector fuel burn model, which can be too cumbersome.

Instead of manually assigning sectors to short-haul and long-haul, we developed a PCA-based clustering method using HDBSCAN as the unsupervised clustering technique. PCA helps reveal the underlying characteristics of different sectors that might not be apparent by considering the raw input factors alone. Boeing 777-300ER sectors are clustered into three, namely short-haul, long-haul AU, and long-haul US and EU. Meanwhile, Airbus 330-300 sectors are clustered into two, namely short-haul and long-haul AU. With a clear characterization of each cluster, one can easily map a new sector to one of the clusters in the absence of data. In this study, only a few sectors were considered due to the limited data shared by our airline partner. When more sectors are involved, the clustering procedure can be more complex and hierarchical clustering may be required if the accuracy of the cluster-based linear regression does not satisfy the requirement.

We showcased the benefits of using a PCA-based clustered model on the estimations of the fuel burn for existing and new sectors. The model is one that balances computational efficiency, effectiveness, and intuitiveness. The parametric nature of the model allows others to use the model even if they have no access to the original datasets used to construct the models, thereby offering wider applicability. The accuracy of the developed PCA-based clustered model was validated by evaluating RMSE and aggregate error, where all RMSE values fell below 5%, and all aggregate errors fell below 2%, including those in fuel burn predictions for new sectors. This predictive capability meets the requirement from our airline partner, i.e., to achieve less than 3% aggregate error. We also demonstrated that the simplified PCA-based clustered model, which only considered three key inputs while assigning nominal values to other factors, still offered sufficient accuracy despite excluding some information. This approach will be practical and beneficial when users do not have access to flight information beyond flight time, zero-fuel weight, and flight distance.

Our new approach shows encouraging results for fuel estimation purposes. Thanks to the data-based nature of our solution method, the same approach can be applied to different datasets corresponding to different aircraft types, fuel types, sectors, airlines, etc. The resulting models can be used for comparison purposes, thereby further enriching our fuel burn estimation studies and providing deeper insight into aircraft fuel performance.

**Author Contributions:** Conceptualization, J.Y. and R.P.L.; data curation, J.Y.; methodology, J.Y.; validation, J.Y.; writing—original draft preparation, J.Y.; writing—review and editing, R.P.L.; supervision, R.P.L. All authors have read and agreed to the published version of the manuscript.

**Funding:** This research received no external funding.

**Data Availability Statement:** Not applicable

**Acknowledgments:** The authors would like to thank Cathay Pacific Airways Limited for providing the data used in this study under the Data Partnership Agreement between the airline and the Department of Mechanical and Aerospace Engineering, HKUST. The authors additionally thank Steve Yip for the discussion and helpful suggestions.

**Conflicts of Interest:** The authors declare no conflict of interest.

## Abbreviations

The following abbreviations are used in this manuscript:

| | |
|---|---|
| HDBSCAN | Hierarchical density-based spatial clustering of applications with noise |
| PCA | Principal component analysis |
| PFIS | Post-flight information system |
| QAR | Quick access recorder |
| RMSE | Root mean square error |
| UMAP | Uniform manifold approximation and projection |

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
