# Peer review of "Cluster-Based Aircraft Fuel Estimation Model for Effective and Efficient Fuel Budgeting on New Routes"

_aerospace, doi:10.3390/aerospace9100624_

Round 1

Reviewer 1 Report

1) The results show that the new proposed model provides the smallest errors in comparison with other models in use today. But if the requirements for the accuracy of assessment exist it will be useful to show and describe. Because it is not understandable - for what we need a more complicated approach? 

2) From fig. 10 one may conclude that the accuracy of the assessment is dependent on the distance of flight for which the assessment is provided. Do the new method essentially influence this accuracy?

3) 'higher errors on longer flights in Airbus 330-300. There are two reasons for this finding. First, there is an imbalance in the distribution of data used to derive the global model... Second, the fuel performance across the different sectors is not homogeneous... it is not surprising that the prediction is less accurate ...' If some recommendations exist to improve - better to show also.

Reviewer 2 Report

Introduction

Very interestingly written with the quotation of several literature items which are examples of fuel consumption models. The literature used on this point is sufficient. Alternatively, I would supplement the content with the reasons for using fuel consumption prediction models for the needs of airlines, e.g. budget, or other issues that force airlines to use such models.

Methodology and results

This chapter provides a good overview of how to create a model that is the work of the authors. I only miss a very clear indication of the differences in the models compared: global model, sector by sector, naivelu-cluster model, PCA-based. In general, the reasons for such large differences were indicated (in particular, Global vs PCA). Is it possible to indicate in more detail?

General remarks and questions

The article is clearly written, the purpose and results of the work are well defined. The results of the model validation are clearly presented. Congratulations on the results of your work. Please, respond to the following questions and comments:

1. Do recently introduced navigation rules such as: Free route aerospace, which mean that a flight on the same route may have a different airway depending on the situation, affect the model?

2. Is it possible to analyze the accuracy of the presented model with others available in the literature? I do not mean to include the analysis in the article, but I am curious whether the information contained in the literature allows for deeper comparisons and analyzes.

3. Is the sentence on line 244 true? Of course, as for the absolute value, yes, but probably not as a percentage.

4. Does the model make it possible to use data from different fuels? The tested planes use aviation kerosene, in the near future in Europe, planes will have a significant share of biofuels. Is it easy to adapt the model to change the fuel consumption coefficients for different flight conditions, or is it just an input data independent of the model?

5. How to explain such a big difference between the models in Figure 10a for HKG-BKK?

6. The airplanes used are about 130t unladen, is the accuracy and sensitivity of the model independent of the aircraft?

Best Rgards!
